# Research on the Interaction Mechanisms between ScCO_2_ and Low-Rank/High-Rank Coal with the ReaxFF-MD Force Field

**DOI:** 10.3390/molecules29133014

**Published:** 2024-06-25

**Authors:** Kui Dong, Shaoqi Kong, Zhiyu Niu, Bingyi Jia

**Affiliations:** 1College of Mining Engineering, Taiyuan University of Technology, Taiyuan 030024, China; dongkui@tyut.edu.cn (K.D.); 15525037025@163.com (Z.N.); 2School of Safety Science and Engineering, Xi’an University of Science and Technology, Xi’an 710000, China; jiabingyiccteg@126.com; 3Xi’an Research Institute of China Coal Technology and Engineering Group Corp., Xi’an 710000, China

**Keywords:** ScCO_2_, low/high-rank coal, coal structure, nano-molecular scale, response mechanism

## Abstract

CO_2_ geological sequestration in coal seams can be carried out to achieve the dual objectives of CO_2_ emission reduction and enhanced coalbed methane production, making it a highly promising carbon capture and storage technology. However, the injection of CO_2_ into coal reservoirs in the form of supercritical fluid (ScCO_2_) leads to complex physicochemical reactions with the coal seam, altering the properties of the coal reservoir and impacting the effectiveness of CO_2_ sequestration and methane production enhancement. In this paper, theoretical calculations based on ReaxFF-MD were conducted to study the interaction mechanism between ScCO_2_ and the macromolecular structures of both low-rank and high-rank coal, to address the limitations of experimental methods. The reaction of ScCO_2_ with low-rank coal and high-rank coal exhibited significant differences. At the swelling stage, the low-rank coal experienced a decrease in aromatic structure and aliphatic structure, and high-rank coal showed an increase in aromatic structure and a decrease in aliphatic structure, while the swelling phenomenon was more pronounced in high-rank coal. At the dissolution stage, low-rank coal was initially decomposed into two secondary molecular fragments, and then these recombined to form a new molecular structure; the aromatic structure increased and the aliphatic structure decreased. In contrast, high-rank coal showed the occurrence of stretches–breakage–movement–reconnection, a reduction in aromatic structure, and an increase in aliphatic structure. The primary reasons for these variations lie in the distinct molecular structure compositions and the properties of ScCO_2_, leading to different reaction pathways of the functional group and aromatic structure. The reaction pathways of functional groups and aromatic structures in coal can be summarized as follows: the breakage of the O–H bond in hydroxyl groups, the breakage of the C–OH bond in carboxyl groups, the transformation of aliphatic structures into smaller hydrocarbon compounds or the formation of long-chain alkenes, and various pathways involving the breakage, rearrangement, and recombination of aromatic structures. In low-rank coal, there is a higher abundance of oxygen-containing functional groups and aliphatic structures. The breakage of O–H and C–OH chemical bonds results in the formation of free radical ions, while some aliphatic structures detach to produce hydrocarbons. Additionally, some of these aliphatic structures combine with carbonyl groups and free radical ions to generate new aromatic structures. Conversely, in high-rank coal, a lower content of oxygen-containing functional groups and aliphatic structures, along with stronger intramolecular forces, results in fewer chemical bond breakages and makes it less conducive to the formation of new aromatic structures. These results elucidate the specific deformations of different chemical groups, offering a molecular-level understanding of the interaction between CO_2_ and coal.

## 1. Introduction

Environmental protection and energy scarcity stand as the two major challenges faced today. The growing awareness of CO_2_ as a greenhouse gas has drawn increasing attention. CO_2_-enhanced coal bed methane (CO_2_-ECBM) projects offer potential environmental benefits by mitigating greenhouse gas emissions through the storage of CO_2_ underground and the promotion of the production of methane [1,2,3]. However, the deployment of ECBM technology necessitates rigorous safety measures to ensure the secure storage of both CO_2_ and CH_4_, addressing concerns such as the prevention of gas leakage, groundwater contamination, and the integrity of geological formations. This can be attributed to the fact that the CO_2_ injected into coal reservoirs exists in a supercritical fluid state (ScCO_2_) [4,5], leading to intricate physicochemical reactions with the coal, resulting in changes in coal structure and properties [6,7,8,9]. The nanoscale pores within coal serve as the primary space for coalbed methane accumulation [10,11]. Alterations in these nanoscale pores can affect the adsorption/desorption capabilities of coal, thereby influencing coalbed methane production and the safety of CO_2_ storage. The interaction between coal and ScCO_2_ is fundamentally a molecular phenomenon [12,13]. The alterations in coal pore structure arise from changes in the molecular structure of coal [14,15]. Therefore, it is essential to conduct nanoscale and molecular coal structure studies to understand the characteristics and mechanisms of the coal structure’s response to ScCO_2_, and thus to provide an effective theoretical basis for the widespread application of CO_2_-ECBM technology.

Currently, scholars have conducted numerous studies on the changes in coal pore structure after ScCO_2_ exposure. Zhang et al. found that after ScCO_2_ treatment, micropores in coal transform into mesopores, leading to a reduction in micropores and an increase in mesopores in different ranks of coal [16]. Chen et al. found that only the mesopore volume increases in medium-rank coal, while both micropore and mesopore volumes increase in high-rank coal [17]. Liu et al. found minimal changes in micropores, with an increase in mesopore volume in both low-rank and high-rank coal, while medium-rank coal exhibited a decrease in mesopore volume [18]. Cheng et al. found that both micropore and mesopore volumes increase in medium-rank coal [19].

The modifications to coal pore structure induced by ScCO_2_ primarily arise from the interaction between ScCO_2_ and coal molecules, resulting in the removal, aggregation, and rearrangement of aromatic structures and functional groups. Wang et al., Sampath et al., and Cao et al. showed that following the interaction of ScCO_2_ with low-rank coal, there is an increase in the content of aromatic structures and oxygen-containing functional groups within the coal, and after its interaction with medium- to high-rank coal, there is a decrease in the content of aromatic structures and oxygen-containing functional groups within the coal [20,21,22]. However, there are also different research conclusions. Wang et al. suggest that, following the interaction with ScCO_2_, there is an increase in aliphatic structures and a decrease in aromatic structures in high-rank coal, with almost no change observed in low-rank coal [23]. Li et al. propose that following the interaction with ScCO_2_, partial aliphatic branches detach in low-rank coal, and oxygen-containing functional groups break, forming more C=C bonds, leading to a decrease in aliphatic content and an increase in aromatic content in low-rank coal [24].

It is evident that the impacts of ScCO_2_ on the changes in pore structure and macromolecular structure in coals of different ranks are not consistent. Experimental methods cannot precisely analyze the essence of ScCO_2_ on coals of different ranks. The reaction mechanism of ScCO_2_ with coals of different ranks should be established through theoretical methods. Currently, methods such as molecular dynamics (MD), grand canonical Monte Carlo (GCMC), and density functional theory (DFT) are widely employed to investigate the adsorption and diffusion characteristics of CO_2_ in coal, as well as the structural changes in coal pores and fractures after CO_2_ injection [25,26,27,28]. The results showed that, after CO_2_ adsorption, not only does the non-covalent bond energy decrease, but changes also occur in bond angle energy, torsion energy, etc., leading to alterations in the macromolecular structure of coal, thereby inducing changes in pore structure [29,30,31,32].

In addition, the reactive force field molecular dynamics (ReaxFF-MD) theoretical method can simulate the in situ reaction process of molecular structures, with a clear identification of its reaction pathways under extreme conditions (such as high temperature and high pressure), coupled with low costs and short processing times. Currently, the ReaxFF-MD method is widely used to investigate the pyrolysis mechanism of coal. Zhang et al. demonstrated that there are three predominant pathways for the initial decomposition of TAGP: the breakage of the C–N bond, the scission of the N–N bond, and the rupture of another N–N bond [33]. Liu et al. showed that the pyrolysis process of low-rank coal was divided into three critical stages: first, the initial pyrolysis process with weak-bridge bonds and the macromolecular network; second, the stage where some tars undergo cracking reactions to generate gaseous products and other tars undergo condensation and polymerization reactions to generate coke; third, the ending of the pyrolysis process [34]. However, there has been no analysis of the interaction mechanism between ScCO_2_ and coal.

Therefore, this study employs the ReaxFF-MD method to calculate the kinetic behavior of low-rank coal (YZ) and high-rank coal (CZ) interacting with ScCO_2_, allowing us to identify the reaction pathways and changes in chemical bonds that occur during the reaction process between coal and ScCO_2_ and analyze the mechanisms of ScCO_2_’s interaction with low-rank coal and high-rank coal. The objective of this research is to provide a theoretical basis to advance the sustainable development of CO_2_-ECBM technology and enhance the efficiency of energy reserves and energy utilization.

## 2. Results and Discussion

### 2.1. The Deformation Characteristics of the Coal Molecular Structure

The changes in coal’s supramolecular structure before and after its interaction with ScCO_2_ are show in Figure 1. At 0–70 ps, coal molecules gradually become less entangled, leading to a decrease in intermolecular distance, while CO_2_ molecules diffuse from the outside to the inside. At 70–200 ps, chemical bonds within the molecules break, leading to the formation of more small radicals (·H, ·OH, ·CH_3_, etc.), and the coal molecular structure gradually diffuses towards the edges. At 200–250 ps, there is no significant alteration in the supramolecular structure of the coal. The main difference between high-rank coal and low-rank coal lies in the degree of molecular structural looseness and the quantity of chemical bonds broken. In high-rank coal, the degree of molecular looseness is greater. In low-rank coal, there is a greater occurrence of chemical bond fractures and small-molecule compounds being generated.

To further analyze the changes in coal structure after ScCO_2_ adsorption, coal macromolecular structures were isolated from the supramolecular structure to observe their deformation characteristics. The coal macromolecular structures before and after the reaction are shown in Figure 2, and the deformation characteristics of coal macromolecular structures at different reaction times are shown in Appendix A. During the reaction process, both low-rank coal and high-rank coal molecules tended to flatten, which is consistent with the results of a previous study on high-volatility bituminous coal [24]. However, molecular simulation methods enable the observation of intermediate processes in the reaction.

For low-rank coal (Figure 2a,b), at 0.00 to 36.00 ps, some thiophene, an oxygen-containing functional group, and aliphatic structures are broken, leading to H^+^ ions being released and the subtle deformation of the molecular structure (Appendix A); from 36 to 48 ps, more aliphatic structures detach from the macromolecular structure, leading to an increase in small-molecule products and the obvious deformation of the molecular structure (Appendix A); from 48 to 69.75 ps, the anthracene in the molecules is broken, leading to the decomposition of the macromolecular structure into two secondary molecular structures (Fragment 1: C_104_H_80_O_7_S; Fragment 2: C_76_H_59_O_7_N_3_) (Appendix A); from 69.24 to 204.00 picoseconds, the two secondary molecular structures continue to react with ScCO_2_ independently (Appendix A); at 204.75 ps, the two secondary molecular fragments recombine into a macromolecular structure (Appendix A); from 204.75 to 250 picoseconds, there is no significant change in the coal molecular structure.

For high-rank coal (Figure 2c,d), from 0 to 31 ps, the aromatic rings connected to oxygen atoms under deformation, and some aliphatic structures transform into aromatic structures due to the dehydrogenation of cyclohexane (Appendix A); from 64.25 ps, aromatic structures in the coal undergo twisting–breakage–movement–reconnection (Appendix A); At 116.25 ps, the flattening of the coal molecular structure reaches its maximum (Appendix A), and by 142.25 ps, the mobile aromatic structures return to their original positions, after which there is no further movement of the aromatic structure within the coal molecule (Appendix A); At 202.25 ps, only a minority of chemical bonds break and form in the molecular structure, with no significant changes occurring.

### 2.2. The Pathways of Functional Groups and Aromatic Structures

The functional groups in coal can be classified into four types: hydrocarbon structures, aliphatic structures, oxygen-containing functional groups, and aromatic structures. The action of CO_2_ on the coal molecular structure can be categorized into two main types: firstly, CO_2_ directly reacts with functional groups; secondly, during the reaction process, H^+^ and CO generated from decomposition react with functional groups. The pathways of functional groups are shown in Figure 3, the pathways of aromatic structures are shown in Figure 4, and details on the reaction pathways are shown in Appendix A.

#### 2.2.1. The Pathways of Functional Groups

For the hydroxyl group pathway (-OH), the CO_2_ attacks the H^+^ atom in the hydroxyl group, inducing a dehydrogenation reaction. The remaining oxygen-containing groups subsequently break, releasing CO and H_2_O molecules (Figure 3: Pathway 1).

For the carbonyl group pathways (-C=O-), the following are observed: (1) hydrogenation, where carbonyl groups combine with free H^+^ ions to form hydroxyl groups (Figure 3: Pathway 2); (2) aromatization, where the O atom in the -C=O- reacts with H^+^, forming an intermediate of C^+^ and ·OH, and C^+^ reacts with a negatively charged methyl ion (CH_3_^−^) and surrounding carbon atoms through an addition reaction, then undergoes rearrangement and dehydrogenation reactions to form the aromatic structure (Figure 3: Pathway 3).

For the carboxyl group pathways (-COOH), the H^+^ attacks the oxygen atom in the carboxyl group (C-OH), causing the detachment of OH- and the formation of a H_2_O molecule. Additionally, the C–C bond breaks, resulting in the formation of CO molecules (Figure 3: Pathway 4).

For the aliphatic structure pathways (-CH_3_), the following occur: (1) decomposition reactions, where the breaking of C_al_–C_al_ or C_al_–H bonds in coal results in the formation of free radical ions such as H^+^ and C^4+^ or smaller hydrocarbon compounds (Figure 3: Pathway 5); (2) a condensation reaction, where the free -CH_3_/-CH_2_ groups detached from the coal macromolecular structure undergo condensation reactions with carbonyl groups or other aliphatic structures in the coal to form long-chain alkenes. From the reaction process diagram, it can be observed that aliphatic structures at the edges of coal macromolecules are more prone to undergoing decomposition reactions, while aliphatic structures connected to aromatic structures or oxygen functional groups are more prone to undergoing addition reactions (Figure 3: Pathway 6).

#### 2.2.2. The Pathways of Aromatic Structures

During the interaction between ScCO_2_ and coal, not all aromatic structures in coal change. Overall, aromatic rings at the edges of macromolecular structures and those connected to side chains are more susceptible to deformation. The changes in aromatic structures mainly include the following forms: (1) aromatic rings being directly broken, gradually changing from cyclic to linear or irregular shapes (Figure 4: Pathway 7); (2) aromatic structures polymerizing with free C^4+^ and H^+^ ions to form C=C bonds (Figure 4: Pathway 8); (3) aromatic rings changing from a six-membered aromatic ring to a seven-membered ring, and then becoming damaged (Figure 4: Pathway 9); (4) aromatic rings first breaking and then recombining during the reaction. At the end of the reaction, there is no change in the aromatic rings (Figure 4: Pathway 10).

#### 2.2.3. The Reaction Pathways of Low-Rank and High-Rank Coal

In low-rank coal (Figure 5), hydroxyl groups (A, B) undergo dehydrogenation (Pathway 1). Aliphatic structures (F, G) aromatize to form aromatic structures (Pathway 6), while aliphatic structures (D, E) decompose to form small hydrocarbon compounds (Pathway 5). The decomposed aliphatic structures combine with carbonyl groups (D, E) to form aromatic structures (Pathway 3). Carbonyl groups (C) undergo substitution reactions only. Carboxyl groups (F, G) break to form carbonyl groups. In aromatic structures, benzene and naphthalene (C, D, F) undergo Pathways 7 and 8, while polycyclic aromatic hydrocarbons (A, B) undergo Pathways 9 and 10.

In high-rank coal (Figure 6), carbonyl groups (③, ④ and ⑦; Pathway 2) undergo hydrogenation to form hydroxyl groups, while a small amount of aliphatic structures (①; Pathway 5) are broken. Similar to low-rank coal, benzene and naphthalene in aromatic structures (②, ③, and ⑤) undergo Pathways 7 and 8. Polycyclic aromatic hydrocarbons (①, ④, ⑥, and ⑦) undergo Pathways 9 and 10.

The molecular fragments located at the edges of the molecular structure and those connected to side chains in both low-rank and high-rank coal are prone to deformation. Compared with high-rank coal, low-rank coal exhibits some differences in reaction pathways. Low-rank coal undergoes more decomposition in terms of its aliphatic structures, resulting in the generation of new aromatic structures. High-rank coal undergoes only a small amount of decomposition in terms of its aliphatic structures, and there is no generation of new aromatic structures.

### 2.3. Chemical Bond Change Characteristics

The changes in the coal molecular structure are primarily caused by the breaking and formation of crosslinks and aromatic rings. Crosslinks in coal can be primarily categorized into C_al_–C_al_ bonds constituting aliphatic side chains, C–O bonds in oxygen-containing functional groups, and C–H bonds. The aromatic structures are primarily composed of C_ar_–C_ar_ bonds.

#### 2.3.1. Low-Rank Coal Chemical Bond Change Characteristics

The curve of chemical bond changes in low-rank and high-rank coal is shown in Figure 7. At 0–48 ps, there is a significant decrease in C–H bonds, C_ar_–C_ar_ bonds rapidly decrease initially and then decrease at a slower rate, C_al_–C_al_ bonds initially increase followed by stabilizing, and a small decrease in C–O bonds occurs. The reduction in aromatic bonds in low-rank coal is mainly due to the breakage of aromatic rings such as thiophene and pyridine. The increase in C_al_–C_al_ bonds is due to the loosening of the molecular structure in low-rank coal, which stretches the C_ar_–C_ar_ bonds in the aromatic ring, transforming them into C_al_–C_al_ bonds.

At 48–69.75 ps, the reduction in C–H bonds accelerates, and there is a slight decrease in C_ar_–C_ar_ bonds and C_al_–C_al_ bonds. This indicates that the coal molecular structure no longer continues to loosen, leading to the fragmentation of aromatic structures. The C–H bonds around the aromatic structure break, forming more free radical ions, and some aromatic structures are decomposed. At 69.25–204 ps, the decrease in C–H bonds continues rapidly, while the reduction in aliphatic structures and the increase in aromatic structures both intensify. This is because during this stage, the condensation effect strengthens, some aliphatic structures transform into aromatic structures, and aromatic structures that had become fragmented in the previous stage recombine. After 204.75 ps, C–H bonds remain essentially unchanged, while the rate of decrease in aliphatic bonds and the rate of increase in aromatic bonds slow down.

#### 2.3.2. High-Rank Coal Chemical Bond Change Characteristics

The curve of chemical bond changes in low-rank and high-rank coal is shown in Figure 8. At 0–31 ps, the increase in C_ar_–C_ar_ bonds and the decrease in C_al_–C_al_ and C–H bonds are attributed to the dehydrogenation of cyclohexane in high-rank coal, resulting in the formation of aromatic structures. Due to the low sulfur content in high-rank coal, more CO_2_ can react with oxygen-containing functional groups in the coal, resulting in a faster decrease in C–O bonds compared with that in low-rank coal.

At 31–116.25 ps, more CO_2_ diffuses into the interior of high-rank coal, leading to a stronger decomposition effect on high-rank coal and the formation of numerous free radical ions. These free radical ions then disrupt the coal molecular structure through chain reactions. As high-rank coal contains a higher proportion of aromatic structures and a lower proportion of aliphatic structures, more aromatic structures are disrupted, leading to the formation of aliphatic structures.

At 116.25–250 ps, although there is movement and reconnection of aromatic structures, the decomposition effect of CO_2_ on coal molecules weakens, and there is no significant change in chemical bonds.

### 2.4. Reaction Mechanism of Low- and High-Rank Coal

Based on the previous discussion, it is evident that there are significant differences in the structural changes, functional group transformation pathways, and chemical bonds broken during the process of high-rank and low-rank coal reacting with ScCO_2_. The main reasons for these differences are related to the reaction mechanisms between ScCO_2_ and coal molecular structures, and the differences in the molecular structures of low- and high-rank coal. The interaction between ScCO_2_ and coal can be divided into two processes: swelling and dissolution.

Stage I: Swelling (Figure 9). During the swelling process, the molecular structure expands, and molecules become less entangled. The main phenomenon involves the breakage of non-covalent bonds between molecules, accompanied by the breaking of weak-bridging bonds within the macromolecular network. Hydrogen bonds, π...π stacking, and the breakage of C_al_–O, C_al_–C_al_, and C_al_–H bonds occur, leading to the dissociation of complex system structures and the release of small-molecule gases. In this process, the molecular center remains unchanged. 

Although ScCO_2_, as a nonpolar solvent, exhibits slightly inferior swelling effects compared to polar solvents, the aromatic structures within coal possess strong adsorption capabilities for nonpolar solvents. Highly aromatic high-rank coals can adsorb more CO_2_. Therefore, the swelling phenomenon of high-rank coal molecules is more pronounced. High-rank coal contains fewer aliphatic structures and aromatic structure stability, with only the partial breaking of C–H bonds in some cyclohexane molecules, leading to the formation of aromatic C_ar_–C_ar_ bonds. Low-rank coal contains more aliphatic structures, and its aromatic structures are unstable. Therefore, during the swelling stage, both C_al_–C_al_ and C_ar_–C_ar_ bonds decrease.

Stage II: Dissolution (Figure 10). In the first stage (Figure 10a), the chain segments of coal macromolecules exhibit enhanced mobility. Due to the internal rotation of the main chain σ bonds, certain segments of the molecule can move relative to others while the molecular center of mass remains unchanged (no plastic deformation occurs). This leads to a reduction in side chains and the fracture of aromatic structures linked to oxygen functional groups in the macromolecular structure of coal. In the second stage (Figure 10b), the movement of chain segments reflects the movement of the entire macromolecular chain (also known as coordinated segmental motion). This stage is primarily characterized by decomposition and condensation reactions, including the breakage of C_ar_–O, C_ar_–C_al_, C_ar_–H, and C_ar_–C_ar_ bonds. During coordinated segmental motion, the impact of ScCO_2_ on the coal molecular structure is most pronounced. The original functional groups and aromatic structures in coal are broken, leading to the detachment of free radical ions, which undergo addition or substitution reactions with the coal macromolecular structure. Due to the weak intra-molecular forces in low-rank coal molecules, they are more prone to breaking and decomposing into smaller molecular units. At this stage, the low-rank coal is decomposed into two second-order molecular fragments, and the aromatic structure of the high-rank coal reflects migration. In the third stage (Figure 10c), under supercritical conditions, along with an increase in molecular collision frequency, coupled with the weak acidity of ScCO_2_, the fragmentation of coal molecules allows them to interact with free radical ions, so the combination reaction occurs, resulting in the formation of new bonds, i.e., molecular structural reorganization. Hence, the phenomenon of molecular recombination in low-rank coal and the reconnection of aromatic structures in high-rank coal are observed in the later stages of the reaction. After the dissolution reaction concludes, both low-rank coal and high-rank coal show no significant changes.

## 3. Simulation Process

The simulation process can be summarized as the selection of YZ [35] and CZ [36] coal molecular structures as research models (Figure 11), analysis of the coal industry, and elemental analysis, as shown in Appendix A. Next, optimization of the coal macromolecular structure model was performed, followed by the construction of supramolecular structures; then, ScCO_2_ adsorbed into the system. Finally, the ReaxFF-MD force field was applied to calculate the changes in small-molecule products, reaction pathways, and chemical bond dynamics following the injection of ScCO_2_. A flowchart is shown in Figure 12.

### 3.1. Simulation Details

#### 3.1.1. Model Construction

(**1**)
**Coal macromolecular structure optimization**


Both the model construction and energy optimization of YZ and CZ molecules were performed using Materials Studio 2019 software. The geometry optimization task in the Forcite module was employed to optimize the structure of the model. The final configuration was an optimized coal model with the lowest energy. Based on energy fluctuations, Metropolis operation rules were employed to either accept or reject changes, thereby facilitating the formation of a new configuration (Figure 12a).

(**2**)
**Construction of Supramolecular Structure Models**


The simulation parameter containing 50 YZ or CZ molecules was built using the Amorphous Cell module. These supermolecule model should be optimized before calculations, and this was performed by using Forcite Module; the simulation parameters were the COMPASS II force field [37], a temperature of 298 K, and the NVT ensemble [38]. The size of the box was 6.77 nm × 6.77 nm × 6.77 nm. The optimized models are shown in Figure 12b.

#### 3.1.2. ScCO_2_ Injection Process

The Adsorption Locator module from Materials Studio was used to place some of the CO_2_ molecules in the supercells, according to the average adsorption under 8 MPa pressure adsorption, as shown in Figure 12. The simulation used the isothermal isobaric ensemble (NPT), selecting Andersen temperature control method (50 °C) and Berendsen pressure control method. The total time period of the simulation was 250 ps, the step size was 1 fs, and the sampling interval was 100 fs. In last 50 ps, data were collected for analysis [39] (Figure 12c).

### 3.2. ReaxFF Force Field Calculate

The ReaxFF force field employs distance-dependent bond order functions to characterize the contribution of chemical bonds to the potential energy. The calculation formula for the original bond order function is shown in Equation (1) [40]:(1)$${BO}_{ij}^{\prime }={BO}_{ij}^{\prime \sigma }+{BO}_{ij}^{\prime \pi }+{BO}_{ij}^{\prime \pi \pi }=exp[p_{bo1}{\left(\frac{r_{ij}}{r_{0}^{\sigma }}\right)}^{p_{bo2}}]+exp[p_{bo3}{\left(\frac{r_{ij}}{r_{0}^{\pi }}\right)}^{p_{bo4}}]+exp[p_{bo5}{\left(\frac{r_{ij}}{r_{0}^{\pi \pi }}\right)}^{p_{bo6}}]$$

In the equation, ${BO}_{ij}^{\prime \sigma }$, ${BO}_{ij}^{\prime \pi }$, and ${BO}_{ij}^{\prime \pi \pi }$ represent the bond orders for single, double, and triple bonds, respectively. $r_{0}^{\sigma }$, $r_{0}^{\pi }$, and $r_{0}^{\pi \pi }$ represent the equilibrium distances for single, double, and triple bonds, respectively. The symbol $p_{bo1}$–$p_{bo6}$ denotes the regression empirical parameters for the ReaxFF force field.

In the ReaxFF force field, the total system energy is represented by Equation (2) [41]:(2)$$E_{system}=E_{bond}+E_{over}+E_{under}+E_{val}+E_{pen}+E_{tors}+E_{conj}+E_{vdWaals}+E_{Coulomb}$$

In the equation, $E_{bond}$ is the bond energy, $E_{over}$ is the over-coordination energy, $E_{under}$ is the under-coordination energy, $E_{val}$ is the bond angle energy, $E_{pen}$ is the loss energy, $E_{tors}$ is the torsional energy, $E_{vdWaals}$ represents the van der Waals forces, and $E_{Coulomb}$ represents the Coulombic forces.

The Reaxff force field in the MAPS program was used to describe the bond formation/breaking and reaction path in the interaction between coal and ScCO_2_ (Figure 12e,f). The NVT ensemble, Berendsen temperature control method, and Mattsson (C/H/O/N/S) reactive force field were adopted in the simulation process.

## 4. Conclusions

(1)The interaction between coal and ScCO_2_ leads to various chemical reactions, including the breakage of aliphatic side chains and oxygen-containing functional group side chains, the removal of heteroatoms, and ring-opening and polymerization reactions between aliphatic and aromatic structures. The process of structural changes in low-rank coal molecules can be summarized as stretches–breakage–recombination, while those in high-rank coal can be summarized as stretches–migration–reconnection.(2)Functional groups and aromatic structures in coal exhibit various reaction pathways. The O–H bond in hydroxyl groups and the C–OH bond in carboxyl groups break. The carbonyl group may undergo hydrogenation to form a hydroxyl group or aromatization with surrounding aliphatic structures to create aromatic ring structures. Aliphatic structures can decompose to form smaller hydrocarbon compounds or condensate to form long-chain alkenes. The reaction pathways of aromatic structures are more complex and involve processes such as breakage, rearrangement, and recombination.(3)The transformation of the coal molecular structure is governed by changes in the chemical bonds within the coal. The content of C–H bonds and C–O bonds decreases in both low-rank and high-rank coals. In low-rank coal, the C_ar_–C_ar_ bonds initially decrease and then increase, while the C_al_-C_al_ bonds initially increase and then decrease. Conversely, in high-rank coal, the C_ar_–C_ar_ bonds initially increase and then decrease, while the C_al_–C_al_ bonds initially decrease and then increase. Overall, in high-rank coal, both C_ar_–C_ar_ and C_al_–C_al_ bonds decrease, while in low-rank coal, C_ar_–C_ar_ bonds increase and C_al_–C_al_ bonds decrease.(4)The responses of high-rank and low-rank coal are related to the structural differences and the interaction mechanisms between ScCO_2_ and coal molecules. Due to the stronger adsorption affinity of aromatic structures for CO_2_, the changes in high-rank coal during the swelling stage are more pronounced compared with those in low-rank coal. At the dissolution stage, chemical bonds in low-rank coal are weaker, the bonds are more prone to breaking after exposure to ScCO_2_, and the disrupted aromatic structures or carbonyl groups combine with detached aliphatic side chains to form larger aromatic structures in low-rank coal. Conversely, in high-rank coal, there are stronger intramolecular forces, with fewer aliphatic structures and carbonyl groups, and new aromatic structures are not formed, leading to a reduction in aromatic structure content.

## Figures and Tables

**Figure 1 molecules-29-03014-f001:**
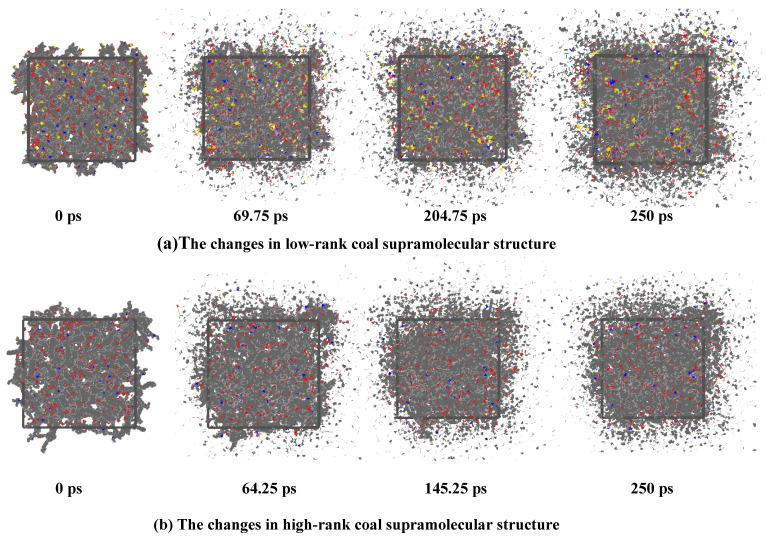
The changes in low/high-rank coal’s supermolecular structure (C: gray; H: white; O: red; S: yellow; N: blue).

**Figure 2 molecules-29-03014-f002:**
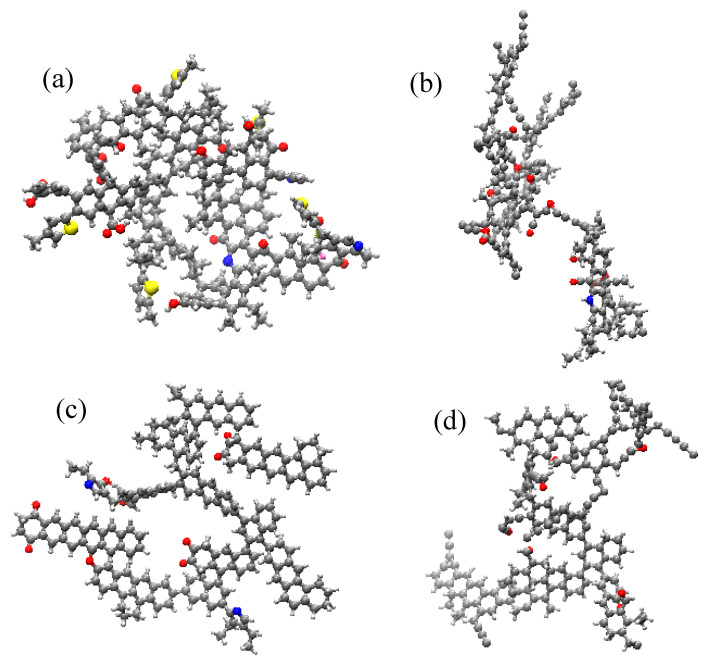
The coal macromolecular structures before and after the reaction with ScCO_2_ ((**a**) the structure of low-rank coal before the reaction; (**b**) the structure of low-rank coal after the reaction; (**c**): the structure of high-rank coal before the reaction; (**d**): the structure of high-rank coal after the reaction) (C: gray; H: white; O: red; S: yellow; N: blue).

**Figure 3 molecules-29-03014-f003:**
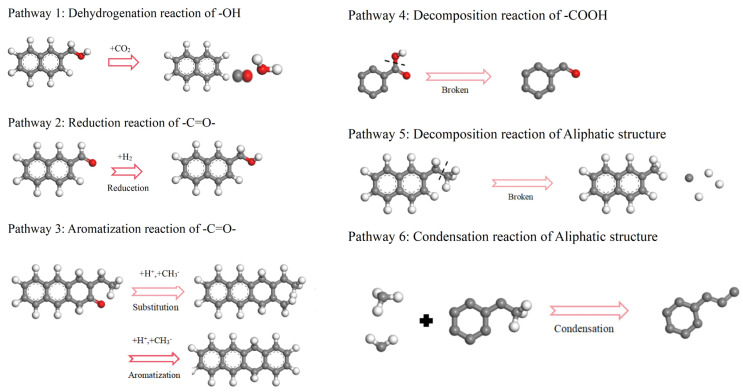
Functional group reaction pathways in coal (C: gray; H: white; O: red).

**Figure 4 molecules-29-03014-f004:**
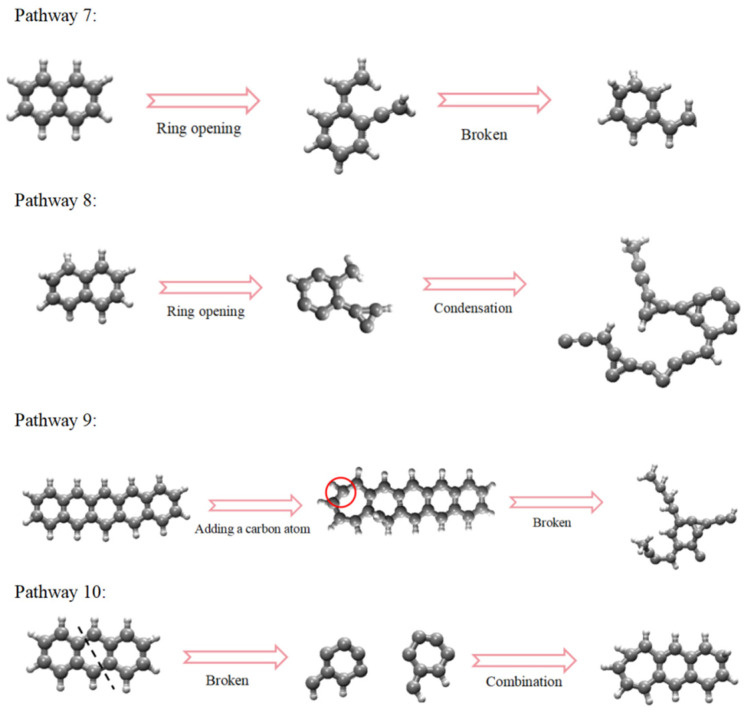
Aromatic structure reaction pathways in coal (C: gray; H: white).

**Figure 5 molecules-29-03014-f005:**
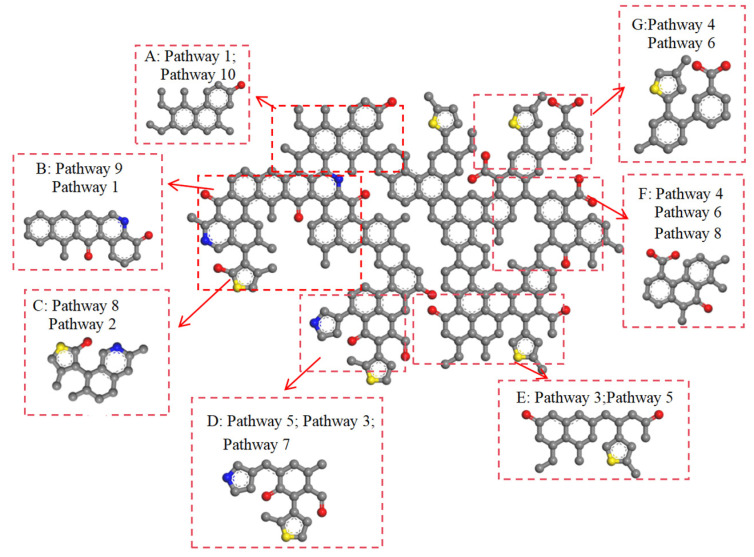
The reaction pathway of low-rank coal (C: gray; H: white; O: red; S: yellow; N: blue).

**Figure 6 molecules-29-03014-f006:**
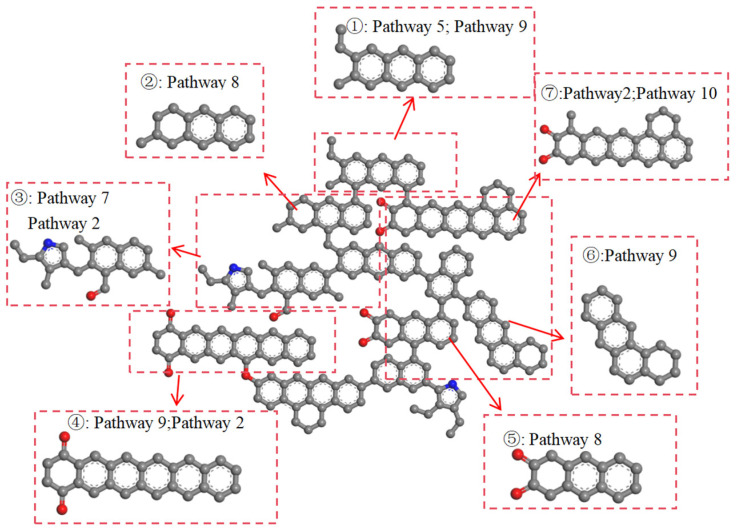
The reaction pathway of high-rank coal (C: gray; H: white; O: red; N: blue).

**Figure 7 molecules-29-03014-f007:**
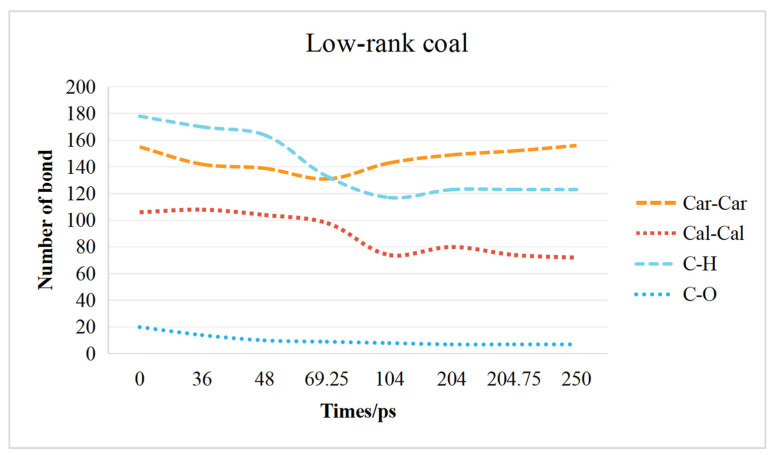
Chemical bond change characteristics of low–rank coal.

**Figure 8 molecules-29-03014-f008:**
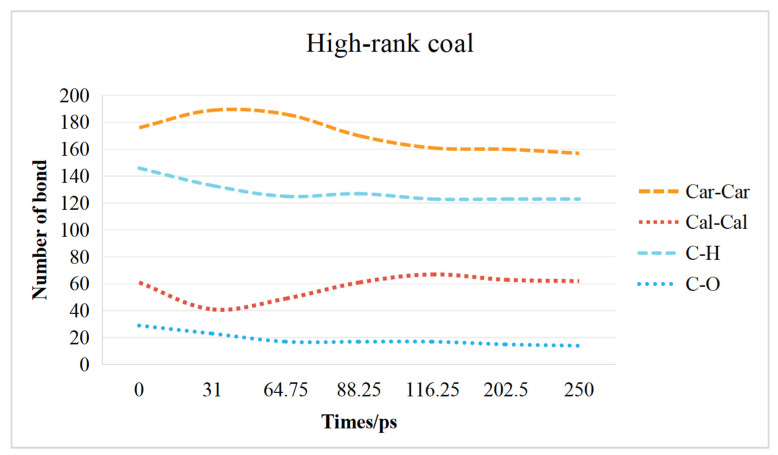
Chemical bond change characteristics of high-rank coal.

**Figure 9 molecules-29-03014-f009:**
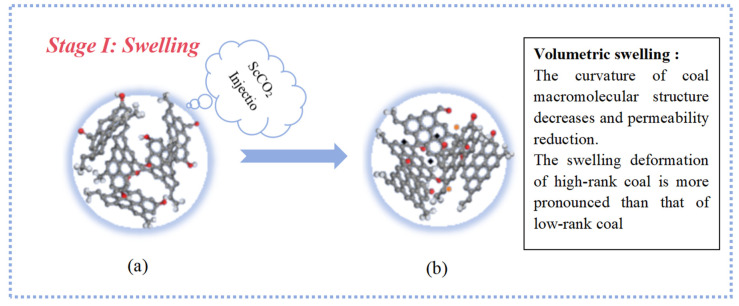
A diagram of the coal swelling process after ScCO_2_ injection (**a**) Coal structure diagram before ScCO2 action; (**b**) Schematic diagram of molecular structure of coal after swelling (C: gray; H: white; The other colors are free radical ions).

**Figure 10 molecules-29-03014-f010:**
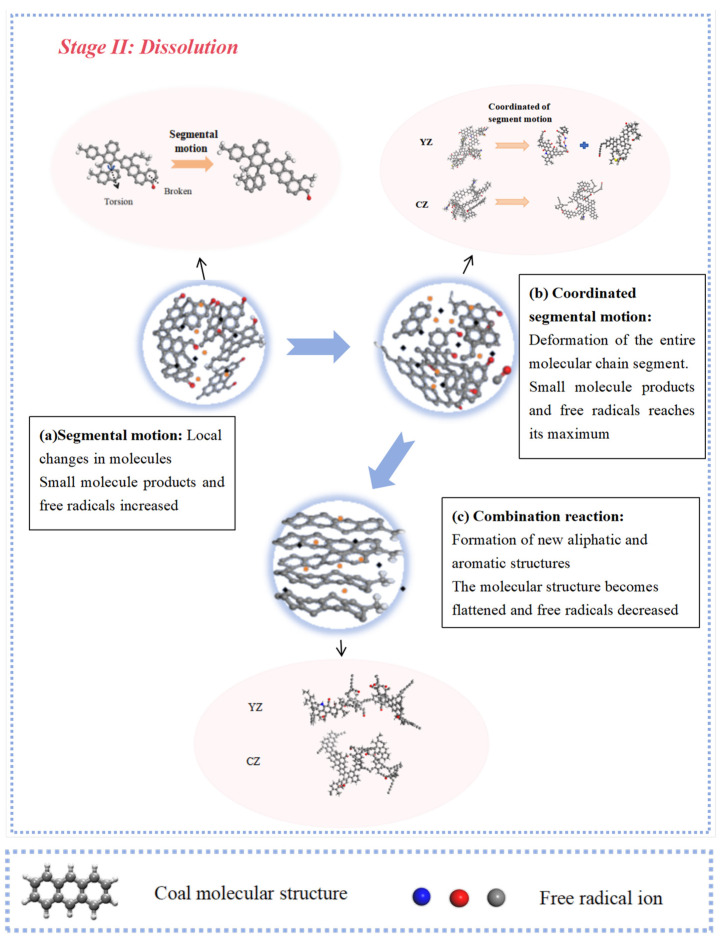
A diagram of the coal dissolution process after ScCO_2_ injection (C: gray; H: white; O: red; N: blue).

**Figure 11 molecules-29-03014-f011:**
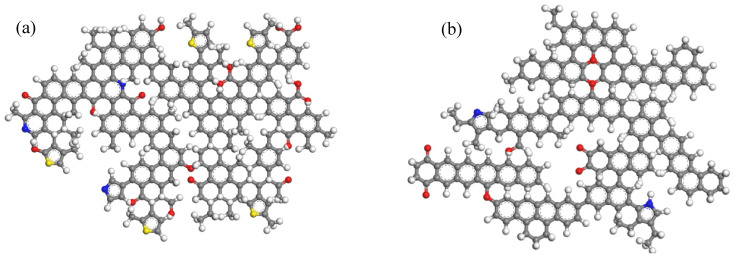
Macromolecular structure of coal: (**a**) low-rank coal, YZ; (**b**) high-rank coal, CZ (C: gray; H: white; O: red; S: yellow; N: blue).

**Figure 12 molecules-29-03014-f012:**
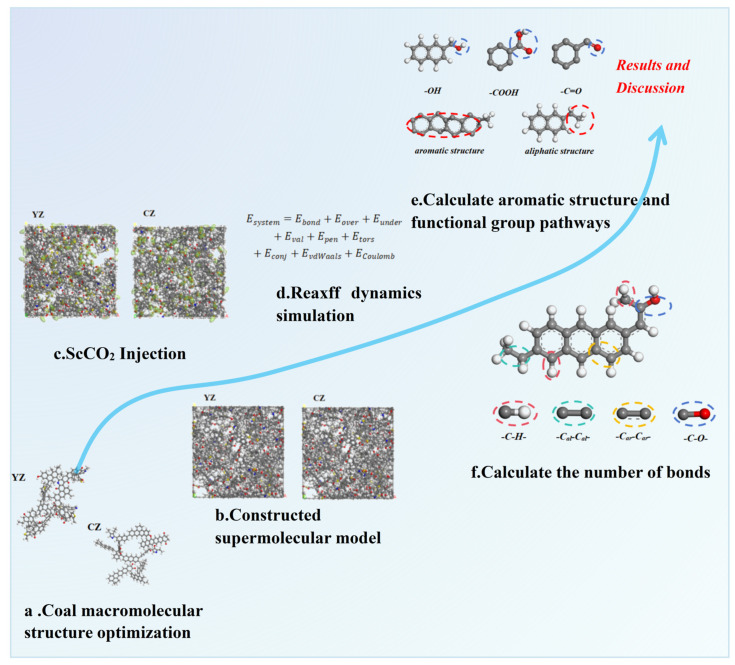
The flowchart of research on the reaction mechanisms between ScCO_2_ and low-rank/high-rank coal (C: gray; H: white; O: red; S: yellow; N: blue).

## Data Availability

Data are contained within the article and Appendix A.

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
