# Peer review of "Research on the Interaction Mechanisms between ScCO2 and Low-Rank/High-Rank Coal with the ReaxFF-MD Force Field"

_molecules, 2024, doi:10.3390/molecules29133014_

Round 1

Reviewer 1 Report

Comments and Suggestions for Authors

I  believe that this paper is scientifically sound and should be published.  It also brings some insight to the molecular mechanisms responsible for the chemical reactions between the ScCO2 and coal as well as physicochemical interactions including coal swelling, pore formation, dissolution of inorganic molecules and extraction of organics. However, the paper does not mention CH4 displacement or any reactions involving CH4 (aspect in passing) or leading to CH4 formation via the generation of .CH3  and .H  radicals.  

However, the most important absence in the paper is the environmental impact of the ScCO2 + Coal and CH4 displacement.   The specific questions that must be answered include:

1.       How safe is CO2 sequestration in coal?

2.       Are there any geological restrictions present for CO2 storage in coal seams?  Restrictions including the type of coal mine?

3.       CO2 storage in coal cannot be regarded as an environmentally friendly process when it releases CH4.  The use of CH4 then defeats the object of the exercise as an environmentally friendly process.  

4.       The process can only be regarded as an equivalent of enhanced oil recovery.  As the concentration of methane is reduced, its recovery will became non-economical and the mine has to be abandoned which can leaded to methane emission.

The authors should  include in “Introduction Section” information regarding the environmental effects of the this technology and safety measures necessary for the safe storage of  CO2 and CH4.  If there are recent peer reviewed papers awailable, these papers should be cited as part of the environmental assessment of the proposed technology.

Reviewer 2 Report

Comments and Suggestions for Authors

This manuscript  conducted ReaxFF-MD simulations to explore the interaction mechanism between ScCO2 and the macromolecular structure of both low-rank and high-rank coal, which provides important insights for guiding coal bed methane projects. Before it can be accepted by molecules journal, some points must be addressed as follows

1. The author used the  ReaxFF-MD method to study the chemical reaction between CO2 and coal molecules, which is a smart way to reveal the realistic environments during CO2 injection, but how to verify the force fields, as we all know that the results are highly dependent on the accuracy of the force field parameter?

2. In figure3, it is hard to identify different groups or molecules. The color code should be provided to better clarify the evolution process.  Besides, why are so many atoms outside the simulation boxes in Figure 3?

3.  In this paper, the author proposed many reaction pathways.Please give some illustrations on these pathways, specifically, how  these reaction pathways are obtained? Based on  ReaxFF-MD results or other methods?

4. Followed by Q3, some pathways seem to be weird, for example, Path3, with the substitution of CO and H, why the final structure doesn't have an O atom? Path9, I don't think the benzene ring will be transformed to a 7-ring structure, do you have any proof about this reaction?

5 Some pathways shown in Figure 7 and 8 are not consistent with Figure 6. For example, path 9, please double check the results and give the explanations. 

Comments on the Quality of English Language

Overall, the expression is fine and I have no more comments on the language part except for line 54-55. Please double check the grammers.

Reviewer 3 Report

Comments and Suggestions for Authors

  • Title: Research on the interaction mechanisms between ScCO2 and low-rank / high-rank coal with ReaxFF-MD force field

    The manuscript provides a comprehensive overview of the research, emphasizing the importance of CO2 geological sequestration in coal seams, the challenges posed by physicochemical reactions between supercritical CO2 (ScCO2) and coal, and the use of ReaxFF-MD theoretical calculations to investigate these interactions. The comparison of low-rank and high-rank coal in terms of their reactivity to ScCO2, both during swelling and dissolution, is well described. The manuscript finishes with a summary of the molecular-level modifications and variances in reaction pathways caused by the coal kinds' different molecular structures. The manuscript makes a substantial and innovative addition to the topic of CO2 sequestration in coal seams. The use of ReaxFF-MD simulations provides valuable insights into the molecular interactions between ScCO2 and coal, resulting in a more complete knowledge that can improve the effectiveness of CCS technology. Addressing the noted flaws, notably the need to clarify technical jargon and include more precise methodological details, will enrich the text and make it more accessible to a wider audience. The work is well-organized and contains important findings. Minor adjustments to improve clarity, include more methodological details and improve result interpretation will have a major impact on the manuscript's overall quality.

Comments on the Quality of English Language

  • Title: Research on the interaction mechanisms between ScCO2 and low-rank / high-rank coal with ReaxFF-MD force field

    The manuscript provides a comprehensive overview of the research, emphasizing the importance of CO2 geological sequestration in coal seams, the challenges posed by physicochemical reactions between supercritical CO2 (ScCO2) and coal, and the use of ReaxFF-MD theoretical calculations to investigate these interactions. The comparison of low-rank and high-rank coal in terms of their reactivity to ScCO2, both during swelling and dissolution, is well described. The manuscript finishes with a summary of the molecular-level modifications and variances in reaction pathways caused by the coal kinds' different molecular structures. The manuscript makes a substantial and innovative addition to the topic of CO2 sequestration in coal seams. The use of ReaxFF-MD simulations provides valuable insights into the molecular interactions between ScCO2 and coal, resulting in a more complete knowledge that can improve the effectiveness of CCS technology. Addressing the noted flaws, notably the need to clarify technical jargon and include more precise methodological details, will enrich the text and make it more accessible to a wider audience. The work is well-organized and contains important findings. Minor adjustments to improve clarity, include more methodological details and improve result interpretation will have a major impact on the manuscript's overall quality.

Round 2

Reviewer 2 Report

Comments and Suggestions for Authors

The author has addressed all the proposed problems and the quality of this work has been improved. One minor question, the author mentioned that they have modified the error in Pathway 3, but in the manuscript it remains the old version, please double check this part.